# Generalized lattices, conformal manifolds, and symmetries

Shlomo S. Razamat⋆, Michal Shemesh and Aelly Zeltzer

Department of Physics, Technion, Haifa, 32000, Israel

⋆ razamat@physics.technion.ac.il

## Abstract

We consider supersymmetric conformal quantum field theories (SCFTs) with degrees of freedom labeled by lattice data. We will assume that in terms of the corresponding lattice the interactions are nearest neighbor and exactly marginal. For example, one can construct such theories by coupling many copies of a single SCFT with exactly marginal deformations. In particular, we discuss the interplay between conformal manifolds of such theories and their global, on-site and lattice, symmetries. We show that one can interpret certain current non-conservation equations for symmetries broken by the interactions as conservation equations including the lattice directions. Moreover, we discuss a class of exactly marginal deformations which are labeled by lattice holonomies that are topological on the lattice. We discuss concrete examples of such constructions and comment on their relevance to compactifications of SCFTs.

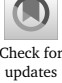

## 1  Introduction

Interacting conformal field theories (CFTs) are defined, in principle, by specifying a set of operators and giving a prescription to determine their correlation functions. Although in lower dimensions this information can be sometimes defined directly, in higher dimensions one typically constructs CFTs as limits of Gaussian, free, field theories. One can start with a free continuous field theory in $D$ space-time dimensions, deform it with relevant (or exactly marginal) interactions and flow to a non trivial CFT in the IR in $D'$ dimensions with $D' \leq D$. Alternatively, one can start from a quantum mechanical system ($D = 1$ QFT) with degrees of freedom labeled by a spatial lattice and consider the limit of taking the size of the lattice to be large and simultaneously focusing on low energy excitations. In this limit sometimes one obtains a continuous $D > 1$ dimensional CFT. We note that in the latter approach some of the continuous space-time symmetries, such as spatial translations and rotations, emerge in the low energy description.

In fact, constructing interacting CFTs with $D > 4$ using field theoretic techniques is a daunting task. Beyond $D = 4$ a deformation of a gaussian fixed point leads to a gaussian fixed point. The shear fact that non-trivial interacting CFTs exist in $D > 4$ is a surprising outcome of string-theoretic constructions [1–4]. Moreover, all well established examples of these constructions require supersymmetry.[1] On the other hand, lattice constructions are harder to analyze once the dimensionality of the spatial lattice is increased.

One can consider constructing higher dimensional CFTs by combining the lattice and the continuum approach. Namely, instead of starting with a quantum mechanical model labeled by lattice data, to start with a $D \geq 1$ dimensional QFT with fields/operators labeled by a lattice. A version of such an approach was discussed in the context of constructing $D > 4$ supersymmetric field theories and goes under the name of deconstruction. To construct a higher dimensional theory one tunes the lattice size to infinity and also some dimensionful parameters (vacuum expectation values (vevs)) in a correlated manner [7]. Similarly, in condensed matter literature constructions of three dimensional theories by coupling together two dimensional CFTs were entertained: this goes under the name of wire constructions (See *e.g.* [8].).

From the modern point of view one can view the deconstruction of the $D = 6$ $(2,0)$ SCFTs as follows. The $D = 4$ lattice theory is obtained by compactifying the $(2,0)$ theory on a torus with punctures (defects) [9]. A good analogy is to consider electrodynamics in presence of atoms (defects) in the low energy limit. In such a limit one often can describe the system as a QM lattice with spin-spin interactions. In deconstruction one then takes the correlated limit of taking the number of defects to infinity and triggering a vev which results in removal of the defects. In the end of this procedure it is believed that the $(2,0)$ on a finite torus (times

---

[1]See a suggestion for a non supersymmetric construction [5,6].

flat $D = 4$ space) is recovered. Note however, that this is rather different from typical lattice constructions in condensed matter physics, where low energy limit leads to some new effective theory, not the QED low energy limit of which in presence of defects we have started with. We will discuss here a construction which bears more resemblances to the more typical condensed matter constructions.

In particular we will consider supersymmetric conformal field theories (SCFTs) with at least four supercharges in $D = 4$ (though most of what we will say holds also for lower dimensional cases with same amount of supersymmetry). SCFTs can possess continuous parameters which do not interfere with conformality. The space of such parameters is complex and is usually called the conformal manifold of the theory.[2] The structure of the conformal manifold has a very tight connection with symmetry properties of the SCFT. In fact following the seminal work of Leigh and Strassler [12] it has been shown that the local structure of the conformal manifold is fixed by a certain Kähler quotient with respect to the complexified global symmetry group [13].[3] We will consider a very specific class of SCFTs which can be constructed by coupling many copies of a basic SCFT together. We will think of the basic SCFT as corresponding to a lattice site and the interactions coupling different sites will be taken to be nearest neighbour: these can be superpotential and/or gauge interactions but will be always assumed to preserve conformality. We will study the interplay of this lattice structure with the symmetry structure of the basic SCFT. For example, before coupling the basic SCFTs together each on-site SCFT has its own copy of global symmetry. Coupling them together will typically break the symmetry to a diagonal combination of all the copies. We will see that the non-conservation of the on-site symmetries by themselves is tightly related to lattice translation. Moreover, some of the exactly marginal deformations can be viewed as topological lattice defect operators for the preserved symmetry.[4]

The fact that the symmetries of an SCFT and their couplings are related has yet another incarnation. Let us start from a $D = 6$ SCFT with some global symmetry $G$ and compactify it on a genus $g$ Riemann surface $\mathcal{C}$. The $D = 6$ theory does not have any continuous parameters [17]. However, upon compactification the $D = 4$ theories generally possess such parameters. The origin of these parameters can be related to symmetries of the original $D = 6$ theory. The $D = 6$ SCFT possesses two special operators, the stress energy tensor and the conserved current multiplets. One can show that upon compactification to $D = 4$ these give rise to exactly marginal deformations [18–20]. For stress energy tensor, which is related to space-time symmetries in $D = 6$, these are related to the complex structure moduli of the compactification surface. For the conserved current multiplet the corresponding exactly marginal deformations are related to the flat connections, holonomies, of the $D = 6$ symmetry group. Many of the lower dimensional SCFTs can be obtained as (deformations of) compactifications of $D = 6$ SCFTs and thus the conformal manifolds of such theories have a deep connections to symmetries of the $D = 6$ theory. We will see how this fact plays out in the lattice constructions.

An interesting question about lattice constructions is whether they possess a simple continuum limit: whether in some limits of the parameters the lattice can be approximated by a vanilla continuum SCFT. Although we will not systematically explore this question here, we will make several comments. First, for such a limit to exist and lead to familiar continuum theories the lattice is expected not to support sub-system symmetries. These are symmetries supported on sub-loci of the lattice and their implications were thoroughly discussed [21, 22] in the context of fractonic phases of matter [23, 24]. It was also observed that sub-system symmetries are common in SCFT constructions [25, 26]. In condensed matter constructions low energy continuum limit is taken by scaling the lattice to be large and possibly taking various

---

[2]This space is also argued to possess Kähler structure [10, 11].
[3]See also [14, 15].
[4]See also [16] for a relation between geometry of the conformal manifold and emergent symmetries.

limits on the couplings. The latter corresponds to taking the lattice spacing to zero. In SCFT the analogues of the lattice spacings will be certain types of exactly marginal deformations. However, the space of exactly marginal coupling of an SCFT often is obtained by a quotient with respect to various duality groups. In particular that might mean that the limit of large coupling ("small lattice spacing") is actually equivalent to small coupling ("large lattice spacing") in another duality frame. It is thus more natural to tune the exactly marginal couplings to self dual loci when attempting to take continuum limits. We will make several brief comments on this issue.

The structure of this paper is as follows. We start in section 2 with a brief review of the relevant facts about conformal manifolds of $\mathcal{N} = 1$ $D = 4$ SCFTs. In section 3 we consider some basic constructions of superconformal one dimensional lattices and discuss the interplay between non-conservation equalities [27, 28] for symmetries and exactly marginal operators. In section 4 we relate some of the properties of the exactly marginal deformations to lattice translations and discuss an interpretation of some of the deformations as symmetry defects. Next, in section 5 we discuss concrete examples of the construction and in 6 we generalize the construction to two dimensional lattices. In particular, we will discuss in section 7 the relation between the lattices and compactifications of $D = 6$ SCFTs to $D = 4$. We summarize and discuss the results in section 8. Several appendices complement the main text of the paper.

## 2 Conformal manifolds and symmetry

We briefly review here the relevant results from [13], notations of which we will follow. Consider an $\mathcal{N} = 1$ theory in $D = 4$ (or a theory in $D < 4$ with supersymmetry being a torus reduction of $D = 4$ $\mathcal{N} = 1$). A general superpotential marginal deformation of an SCFT takes the form

$$W = \sum_i \lambda_i \cdot \mathcal{O}^i \,. \tag{1}$$

The operators $\mathcal{O}^i$ form representation $\mathcal{R}$ of global symmetry $G$ of the SCFT. The deformation breaks the symmetry $G$ and we can write

$$\overline{\mathrm{D}}^2 J^a(\mathbf{x}) = \sum_i X_i^a(\{\lambda\}) \mathcal{O}^i \,, \qquad X_i^a(\{\lambda\}) = \lambda_j (T^a)^j{}_i + O(\lambda^2) \,. \tag{2}$$

Here $(T^a)^j{}_i$ are the representation $\mathcal{R}$ matrices of generators of $G$. The quantity $X_i^a$ is a vector field on the space of couplings representing the action of the group $G$ on it. Let us mention here that $\mathrm{Im} \int d^2\theta \left( \overline{\mathrm{D}}^2 J^a(\mathbf{x}) \right) \propto \partial^\mu j_\mu^a$. The one loop computation of the $\beta$-function implies that for small $\lambda_i$ the deformation is either marginally irrelevant or exactly marginal. It is exactly marginal if

$$D^a \equiv 2\pi^2 \lambda_i (T^a)^{ij} \overline{\lambda}_j + \cdots = 0 \,. \tag{3}$$

For exactly marginal deformation to exist, it is enough to find a solution to the equation above keeping only the leading terms. This solution can always be corrected with higher order terms in $\lambda_i$. Moreover, we should identify the solutions related by the complexified symmetry group $G_{\mathbb{C}}$ and thus can write that the manifold of exactly marginal deformations, *the conformal manifold $\mathcal{M}_c$*, is given by,

$$\mathcal{M}_c = \{\lambda_i\}/G_{\mathbb{C}} \,. \tag{4}$$

The addition of gauge interactions to this discussion is straightforward. The gauge interactions add a gauge coupling in $D = 4$ as a marginal parameter. This coupling can be charged

under an anomalous U(1) symmetry. Thus to compute the Kähler quotient we simply list also all the gauge couplings and quotient also by complexified anomalous U(1) symmetries. The computation of this quotient might be non trivial. For a detailed algorithm to compute the quotient and various subtleties associated with it one can consult [29]. See also [30–35] for additional discussions and examples.

# 3 From non-conservation to conservation on the lattice

Let us consider an example of a conformal field theory which can be thought of as being associated to a lattice. This example can be easily generalized in several ways.

Let us consider a supersymmetric CFT, $\mathcal{T}$, with a U(1) global symmetry. We assume that the theory has operators with superconformal R-charge [36] 1 which are charged $\pm 1$ under the U(1) symmetry, $\mathcal{O}^{\pm}$. We will also assume that the theory is invariant under charge conjugation $\mathcal{C}$ exchanging the positive and negative charges. That is we assume the symmetry is actually $U(1) \rtimes \mathcal{C} = O(2)$. The discussion here will assume only these properties. We will later discuss a concrete theory realizing this particular setup. The discussion here can be done for theories in $1 < D \leq 4$.

Next, we consider taking $L$ copies of the theory above, $\mathcal{T}_i$, $i = 1 \cdots L$. Each such theory has its own O(2) symmetry. We note that taking two theories and coupling them with the superpotential,

$$W = \lambda (\mathcal{O}_1^+ \mathcal{O}_2^- + \mathcal{O}_1^- \mathcal{O}_2^+), \qquad (5)$$

is an exactly marginal deformation preserving the diagonal O(2) symmetry of the two theories. We have two O(2) symmetries to start with and two marginal deformations, $\mathcal{O}_1^+ \mathcal{O}_2^-$ and $\mathcal{O}_1^- \mathcal{O}_2^+$. However, only turning both deformations together gives an exactly marginal deformations while the orthogonal combination recombines with a combination of the conserved currents of the two U(1) symmetries. Note that we could choose to turn on a relative phase between the two terms in the superpotential and still obtain an exactly marginal deformation. Such a phase amounts to acting with a global symmetry and all such choices are equivalent. However, as it will be important later on, we stress that we have exhausted this freedom of choice here.

Next, we consider coupling the $L$ copies of the theory together through a superpotential,[5]

$$W = \sum_{j=1}^{L} \lambda_j (\mathcal{O}_j^+ \mathcal{O}_{j+1}^- + \mathcal{O}_j^- \mathcal{O}_{j+1}^+), \qquad (6)$$

with $\mathcal{O}_{L+1}^{\pm} = \mathcal{O}_1^{\pm}$. This coupling is exactly marginal and preserves an O(2) diagonal subgroup of the global symmetry. We can think of the resulting theory as a one dimensional periodic lattice of QFTs sitting at the nodes. Let us denote this theory by $\mathcal{T}^{(L)}$.

In case that all the couplings are the same, $\lambda_i = \lambda$, in addition to the O(2) symmetry the theory has also a $\mathbb{Z}_L$ translation symmetry $T$. Taking a general operator $\mathcal{O} \in \mathcal{T}^{(L)}$ built from operators of $\mathcal{T}_i$ the symmetry $T$ acts by shifting the indices of each constituent operator $i \to i + 1$,

$$T \mathcal{O} T^{-1} = \mathcal{O}(i \to i + 1), \qquad T^L = 1. \qquad (7)$$

---

[5]Note that in principle in such examples we can also discuss exactly marginal deformations which are not nearest neighbor and also the theories $\mathcal{T}_i$ might have on-site exactly marginal deformations. However we will refrain from doing so. Note also that superpotential deformations can be exactly marginal only in strongly coupled SCFTs. In weak coupling for a superpotential deformation to be exactly marginal one needs to have also gauge interactions [13].

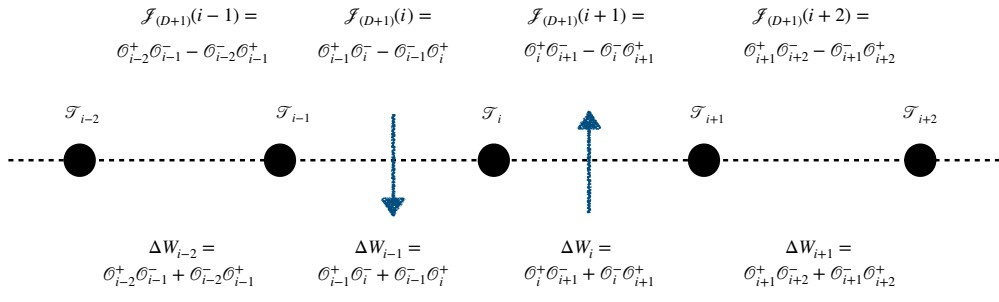

Figure 1: A one dimensional lattice. For each lattice site $i$ we have a copy of the basic theory $\mathcal{T}_i$. For each edge of the lattice we have a term in the superpotential coupling two neighbouring copies. We can associate to the edges a topological defect operator (depicted here by the vertical blue arrows) which is exactly marginal. The arrow indicates the sign of the contribution of the current to the non-conservation equation.

Note that there are many more exactly marginal deformations that the theory $\mathcal{T}^{(L)}$ has. We next turn to analyze some of these deformations and their relation to the symmetries.

Note that before we couple the theories together each one has a copy of symmetry $G = O(2)$ with an accompanying conserved current which sits in a linear supermultiplet $J_i$. The conservation equations read

$$\forall i = 1 \cdots L, \qquad \overline{D}^2 J_i(\mathbf{x}) = 0. \tag{8}$$

However, coupling the theories together the symmetry is broken to the diagonal one. The non-conservation equation becomes

$$\overline{D}^2 J_i(\mathbf{x}) = X(\lambda) \left[ \left( \mathcal{O}_i^+ \mathcal{O}_{i+1}^- - \mathcal{O}_i^- \mathcal{O}_{i+1}^+ \right) - \left( \mathcal{O}_{i-1}^+ \mathcal{O}_i^- - \mathcal{O}_{i-1}^- \mathcal{O}_i^+ \right) \right]. \tag{9}$$

Here $X(\lambda) = \lambda + O(\lambda^2)$ if all the couplings are small [13]. This is an operatorial equation. As usual, if one has additional insertions in the correlation function, one should add appropriate contact terms on the right hand side. These will become important later on. Now note that if we define

$$\mathcal{J}_{(D+1)}(\mathbf{x}, i) \equiv X(\lambda) \left( \mathcal{O}_{i-1}^- \mathcal{O}_i^+ - \mathcal{O}_{i-1}^+ \mathcal{O}_i^- \right), \tag{10}$$

the non conservation can be written as

$$\overline{D}^2 J_i(\mathbf{x}) + \mathcal{J}_{(D+1)}(\mathbf{x}, i+1) - \mathcal{J}_{(D+1)}(\mathbf{x}, i) = 0, \tag{11}$$

which has the structure of a conservation equation in $D + 1$ dimensions with the extra dimension discretized. In particular, the usual conservation of currents appears in the (imaginary part of the) F-terms of the equation above and we can write

$$\partial_\mu j^\mu(\mathbf{x}, i) + j^{(D+1)}(\mathbf{x}, i+1) - j^{(D+1)}(\mathbf{x}, i) = 0, \tag{12}$$

where we have taken the F-term of (11) and defined,

$$j^{(D+1)}(\mathbf{x}, i) \propto \int d^2\theta \, \mathcal{J}_{(D+1)}(\mathbf{x}, i) - \text{c.c.} \tag{13}$$

The non-conservation relation implies that the marginal operators recombine with the currents of the broken symmetry and become non-chiral: these marginal operators and the broken currents develop the same anomalous dimensions. In other words these marginal operators become marginally irrelevant [13, 14]. We expect this anomalous dimension to grow as we increase the coupling $\lambda$. We turned on lattice independent couplings and thus have a lattice translation symmetry. The $L$ marginally irrelevant operators are associated to the lattice positions and have same charges under continuous global symmetries.[6] In general thus these operators can mix. However, because of the lattice translation symmetry we expect to be able to diagonalize the mixing in the lattice momentum basis. We thus in general expect the anomalous dimensions of the marginally irrelevant operators to occupy a "band", the width and details of which might depend on the coupling $\lambda$. Moreover, all the other unprotected operators also should form such "bands". Some of the protected operators will not acquire anomalous dimensions and will occupy "bands" of zero width.[7]

Let us next consider the conserved charges. The conserved charge of the diagonal symmetry is

$$Q^0(t) = \sum_{i=1}^{L} \int d^{D-1}\mathbf{x}\, j^0(\mathbf{x}, i), \tag{14}$$

and is an integral over the continuous $D-1$ dimensional space as well as a summation over the discretized lattice direction. Whenever we have a symmetry in addition to charges, which are associated to surfaces localized in time and extended in space, one can consider defects which are associated to surfaces localized in one of the spatial directions and extending in time. First, we can consider defects localized in one of the continuous directions,

$$\mathcal{Q}^\mu(x^\mu) = \sum_{i=1}^{L} \int \left( \prod_{\nu \neq \mu} d\mathbf{x}_\nu \right) j^\mu(\mathbf{x}, i), \tag{15}$$

which can be again though of as an integral over time, $D-2$ spatial directions and the discrete direction. Second, we can localize them on the lattice and obtain the following,

$$\begin{aligned} \mathcal{Q}^{(D+1)}(i) &= \int d^D\mathbf{x}\, j^{(D+1)}(\mathbf{x}, i) = \int d^2\theta \int d^D\mathbf{x}\, \mathcal{J}^{(D+1)}(\mathbf{x}, i) - \text{c.c.} \\ &\propto \int d^2\theta \int d^D\mathbf{x} \left( \mathcal{O}_{i-1}^- \mathcal{O}_i^+ - \mathcal{O}_{i-1}^+ \mathcal{O}_i^- \right) - \text{c.c.} \end{aligned} \tag{16}$$

Because of equation (11), $\mathcal{Q}^{(D+1)}(i)$ and $\mathcal{Q}^{(D+1)}(j)$ with $i \neq j$ differ by irrelevant operators at most. The operator $\mathcal{Q}^{(D+1)}(i)$ can be added to the action and corresponds to a certain marginal superpotential deformation. Moreover, this deformation is *exactly marginal* (plus possibly marginally irrelevant deformations). One way to see this is that once we turn on a non zero $\lambda$ this operator does not break any continuous symmetries of the theory and thus has to be exactly marginal [13]. It breaks however the $\mathbb{Z}_2$ symmetry.[8] This operator is naturally associated to the link between $i$th and $(i-1)$th nodes of the lattice. As this deformation is independent of the lattice location we can think of it as being *topological on the lattice*. To be more precise placing the deformation (16) at different lattice points are related by marginally irrelevant deformations. See Figure 1 for an illustration. As we will see in the next section,

---

[6]More generally there might exist symmetries which are "localized" on sites/sub-lattices which will forbid/restrict the mixing. Examples of these are sub-system symmetries discussed in this context *e.g.* in [25].

[7]We thank Z. Komargodski for suggesting the band interpretation of the spectrum of anomalous dimensions.

[8]Note that the relative phase between the two terms in the deformation is material as we have already used the freedom to rotate the operators by a phase writing the $\lambda$ deformation.

placing the deformations at different locations on the lattice with couplings satisfying certain property would produce same correlation functions up to an action of the symmetry generators on some of the operators with parameters depending on the couplings.

Let us make several further comments. The deformations we have discussed can be turned on with infinitesimal couplings. Once we will turn on finite couplings the exact form of the deformation might change.[9] We will address this issue more in the next section.

Let us perform a simple counting of nearest neighbor exactly marginal deformations preserving the U(1) symmetry. We have in total $2 \times L$ marginal deformations, $L$ deformations $\mathcal{O}_{i-1}^{+}\mathcal{O}_{i}^{-}$ and $L$ deformations $\mathcal{O}_{i-1}^{-}\mathcal{O}_{i}^{+}$, and $L$ U(1) symmetries. The preserved U(1) is the diagonal combination of all the U(1)s. Thus the number of (nearest neighbor) exactly marginal deformations is $2 \times L - L + 1 = L + 1$. The $L$ deformations correspond to the links and the $+1$ is the topological deformation.[10]

If we are to consider an open chain of theories then the counting of the exactly marginal deformations is slightly modified. If we have an open gluing of $L$ theories, we have $L$ U(1) symmetries but only $2 \times (L-1)$ interactions of the type we consider. This leads to $2 \times (L-1) + 1 = L - 1$ exactly marginal deformations. These correspond to the links and we do not have the topological deformation. If we choose to close the chain into a ring we do not break any symmetries but add two deformations. Thus the topological deformation is really associated to the holonomy around the circle. Note, conversely, that it does not matter where we glue a chain into the ring and that is why the additional deformation is not associated with a particular position on the ring. Finally, we do not have a translation symmetry on the chain as it is broken by the boundaries. As we will see the topological operator has an interesting connection with the translation symmetry of the ring.

As deformations corresponding to different links differ by marginally irrelevant operators, we can wonder how to build an exactly marginal deformation without an admixture of irrelevant ones. A natural candidate for this, dictated by symmetry, is to "smear" the deformation on the lattice: turn on all the $\mathcal{Q}^{(D+1)}(i)$ with same coefficients.

Let us comment on how these considerations are reflected in the various supersymmetric partition functions one can consider. For example, the supersymmetric index [38] is independent of the exactly marginal deformations but it can be refined by fugacities for any global symmetry. In particular, the setup we are discussing has a $\mathbb{Z}_L$ translation symmetry and thus the index of the theory can be refined with a fugacity for a generator of this symmetry, $a$ ($a^L = 1$). The $L$ exactly marginal deformations corresponding to the links can be grouped into the $L$ different one dimensional irreps of $\mathbb{Z}_L$.[11] The topological deformation, as it is not associated to any position, will give rise to an additional $\mathbb{Z}_L$ singlet.

---

[9]One can study the exactly marginal deformations of the theory deformed by the infinitesimal deformations we discussed. The deformed theory will have different non-conservation equations. Leading to a correction to the deformation in higher orders.

[10]We can view the couplings corresponding to the links as defining "lengths" for the links. Naturally one can think of the zero coupling limit as an infinite length limit. We can also think of these deformations defining a non-trivial metric on the circle. An interesting algebraic structure related to this type of deformations in a particular example where the lattice structure is related to an orbifold was discussed recently in [37].

[11]That is we can define operators which are irreps under the translation symmetry,

$$\hat{\mathcal{J}}_{(D+1)}^{n}(\mathbf{x}) \equiv \sum_{k=1}^{L} e^{\frac{2\pi i k n}{L}} \mathcal{J}_{(D+1)}(\mathbf{x}, k),$$

which are just operators with well defined lattice momentum.

# 4 Generalized lattices and symmetries

Next we will connect the topological exactly marginal deformation we found to a defect on the lattice for the U(1) symmetry. Our discussion will parallel closely similar considerations in the case of quantum mechanical lattice models [39]. Let us consider the following marginal deformation,

$$\Delta W_1 = \lambda \left( e^{-i\theta} \mathcal{O}_L^+ \mathcal{O}_1^- + e^{i\theta} \mathcal{O}_L^- \mathcal{O}_1^+ \right) - \lambda \left( \mathcal{O}_L^+ \mathcal{O}_1^- + \mathcal{O}_L^- \mathcal{O}_1^+ \right). \tag{17}$$

This deformation changes the coupling on the link between the $L$th and the 1st sites. To linear order in $\theta$ it coincides with (16) and thus is exactly marginal. We will next want to claim that the above is the correct exactly marginal deformation one needs to add to the action even for finite $\theta$. Although one can always complexify the exactly marginal parameters, for the argument below to hold we will take importantly $\theta$ to be real.[12]

Note that the deformation is equivalent to changing the superpotential on the link between $L$th node and node 1. The deformation with $\theta$ breaks the charge conjugation $\mathbb{Z}_2$ symmetry but preserves the U(1). Moreover, the translation operator $T$ (7) does not generate a symmetry anymore. However, we still have a translation symmetry acting as follows. We consider the action of translation $T$ together with the phase rotation of the U(1) of theory $\mathcal{T}_1$, $R_1(\theta)$. Under this transformation,

$$R_1(\theta) T (W + \Delta W_1) T^{-1} R_1(\theta)^{-1} = W + \Delta W_1, \tag{18}$$

and thus this is a symmetry of the theory.[13] See next sub-section for a simple proof. Note that $R_1(\theta)$ by itself is not a symmetry of the theory and $T$ is not a symmetry but together they combine to a symmetry. Moreover,

$$(R_1(\theta) T)^L = \prod_{i=1}^{L} R_i(\theta) = R(\theta), \tag{19}$$

with $R(\theta)$ being the rotation of the preserved U(1) symmetry of the theory. We learn that this exactly marginal deformation corresponds to a very particular deformation of the symmetry $U(1) \times \mathbb{Z}_L$ determined by $\theta$. We can repeat this procedure with any link of the lattice always preserving some extension of the symmetry. Let us denote the new translational symmetry as

$$\widetilde{T}(i, \theta) = R_i(\theta) T. \tag{20}$$

Because the deformed theory possesses the discrete symmetry which depends on $\theta$ we expect this deformation, even for finite $\theta$, to be exactly marginal.

Let us assume now that we compute a correlation function of a theory deformed by $\theta$ with $n$ operators which are local on the lattice, $\mathcal{O}(\mathbf{x}_\ell, i_\ell)$ with $i_\ell$ the lattice location of the $\ell$th operator. Let us assume that $i_\ell \neq L \ \forall \ell$. Now, we can act with symmetry $\widetilde{T}$. On $\mathcal{O}(\mathbf{x}_\ell, i_\ell)$ when $i_\ell \neq L$ it will act as

$$\widetilde{T} \mathcal{O}(\mathbf{x}_\ell, i_\ell) \widetilde{T}^{-1} = \mathcal{O}(\mathbf{x}_\ell, i_\ell + 1), \tag{21}$$

*i.e.* as a usual translation. However, if $i_\ell = L$,

$$\widetilde{T} \mathcal{O}(\mathbf{x}_\ell, L) \widetilde{T}^{-1} = e^{i\theta q_{\mathcal{O}}} \mathcal{O}(\mathbf{x}_\ell, 1), \tag{22}$$

---

[12]Our argument is based on symmetry considerations. Although the superpotential is invariant under complexified transformations the Kähler potential is not.

[13]Note that we start with a Kähler potential which does not couple different copies of $\mathcal{T}$ and thus is also invariant under this symmetry.

where $q_{\mathcal{O}}$ is the charge of the operator. One way to understand the phase is to remember that the non-conservation equation (9) is modified in presence of operators in the expectation value by contact terms. We can think of this picture as follows. On one hand the transformation keeps the deformation unchanged: that is why it is a symmetry of the deformed action. However, it translates all the operators. If an operator crosses the lattice location of the deformation it acquires a phase depending on its charge. Alternatively, we can think of this as translating the deformation. Once we pull it through an operator we acquire a phase. Note that it is important here that the deformations are associated to the edges, *i.e.* to the dual lattice.

Note that with this understanding we can consider two deformations at two different sites and repeat the analysis. This will lead to symmetry generated by $\widetilde{T}(i, \theta_1, \theta_2) = R_{i_1}(\theta_1) R_{i_2}(\theta_2) T$ or in more generality to,

$$\widetilde{T}(\{\theta_i\}) = \left(\prod_{i=1}^{L} R_i(\theta_i)\right) T\,. \tag{23}$$

We can repeat the argument with the operators acquiring a phase when pulling through one of the locations where non-trivial topological operator resides. However, we can also say something stronger. Let us consider two deformations at two different locations, say $i$ and $j$. We consider turning on first the deformation at $i$ and having the corresponding twisted symmetry $\widetilde{T}(\theta_i)$. We then think of turning on the deformation at $j$ in conformal perturbation theory around the deformation at $i$. In particular we can act with $\widetilde{T}(\theta_i)$ which should be a symmetry of the theory. This transformation keeps the deformation at $i$ in place but moves the deformation at $j$. As the deformations are singlets of the group they will not feel each other and also their relative position is inessential. We can move the deformations around and merge them as long as we do not cross charged operators. If we do, the results are multiplied by a phase.

Yet another way to understand the topological nature of the deformation is as follows. We are free to redefine the operators by acting with the broken U(1) symmetries. For example, smearing the deformation over the whole lattice with equal $\theta$s we can remove the deformation on one of the links, say $L-1$, by a U(1) transformation on site 1. This will however change the deformation on link $1-2$ to $2\theta$. We can repeat this procedure in many ways, and in particular we can move the deformations to a single link to be $L\theta$. However, we cannot remove the deformation completely.[14]

We can consider a different deformation, preserving the parity $\mathcal{C}$ but breaking the U(1) symmetry. We consider the following deformation,

$$\Delta W_2 = \lambda(\mathcal{O}_L^+ \mathcal{O}_1^+ + \mathcal{O}_L^- \mathcal{O}_1^-) - \lambda(\mathcal{O}_L^+ \mathcal{O}_1^- + \mathcal{O}_L^- \mathcal{O}_1^+)\,. \tag{24}$$

Note that this deformation does not have a continuous parameter associated to it. Again we have explicitly broken the translation symmetry generated by $T$ but a different symmetry emerges

$$\mathcal{C}_1 T (W + \Delta W_2) T^{-1} \mathcal{C}_1 = W + \Delta W_2\,, \tag{25}$$

with $\mathcal{C}_1$ parity transformation of $\mathcal{T}_1$. The new transformation satisfies

$$(\mathcal{C}_1 T)^L = \prod_{i=1}^{L} \mathcal{C}_i = \mathcal{C}\,. \tag{26}$$

---

[14]Note that this way of understanding the topological nature of the deformation has the advantage that it makes no reference to the translation symmetry. In particular it will hold if we turn on different $\lambda$ couplings for the links. On the other hand if we cut the lattice cycle open, performing redefinitions of operators with broken symmetries can remove the deformation we are discussing here by moving them all "beyond" the boundary.

Thus again the symmetry $\mathbb{Z}_L \times \mathcal{C}$ is extended. In fact the symmetry becomes just $\mathbb{Z}_{2L}$ generated by $\mathcal{C}_1 T$. We have one independent deformations of this sort as it is topological on the lattice.[15]

## 4.1 More general superpotential construction

We can generalize the construction in various ways. We assume that theory $\mathcal{T}$ has marginal operators of R-charge one, $\mathcal{O}_i$, in representations $\mathcal{R}_i$ of the symmetry group $G$. We consider gluing the two theories with the marginal superpotential $W = \sum_{i \neq j=1}^n \lambda_{ij} \mathcal{O}_i^{(1)} \mathcal{O}_j^{(2)}$ and assume that the Kähler quotient $\{\lambda_{ij}\}/G_{\mathbb{C}}^{(1)} \times G_{\mathbb{C}}^{(2)} \neq \emptyset$. In such a case the analysis of the previous section can be repeated verbatim. Instead of performing the most general analysis we will focus on a useful special case.

We specialize the construction above to any group $G$ and the operators $\mathcal{O}^{(R)}$ and $\mathcal{O}^{(\overline{R})}$, with $R$ and $\overline{R}$ conjugate representations of $G$. The Kähler quotient is not empty. We consider the lattice,

$$W = \lambda \sum_{j=1}^L \left( \mathcal{O}_j^{(R)} \cdot \mathcal{O}_{j+1}^{(\overline{R})} + \mathcal{O}_j^{(\overline{R})} \cdot \mathcal{O}_{j+1}^{(R)} \right). \tag{27}$$

The product $\cdot$ projects on a $G$ singlet in the tensor product of $R$ and $\overline{R}$. The theory has a translation symmetry $T$ as before. We consider a deformation labeled by a group element $g \in G$ and a lattice position (chosen here to be 1 for concreteness),

$$\Delta W_g^1 = \lambda \left( \mathcal{O}_L^{(R)} \cdot g \left( \mathcal{O}_1^{(\overline{R})} \right) + \mathcal{O}_L^{(\overline{R})} \cdot g \left( \mathcal{O}_1^{(R)} \right) \right) - \lambda \left( \mathcal{O}_L^{(R)} \cdot \mathcal{O}_1^{(\overline{R})} + \mathcal{O}_L^{(\overline{R})} \cdot \mathcal{O}_1^{(R)} \right). \tag{28}$$

Note that this is an exactly marginal deformation. The subgroup preserved by this deformation is the centralizer of $g$ in $G$. We claim that

$$g_1 T (W + \Delta W_g^1) T^{-1} g_1^{-1} = W + \Delta W_g^1. \tag{29}$$

To show this we compute as follows,

$$
\begin{aligned}
\lambda^{-1} g_1 T (W + \Delta W_g^1) T^{-1} g_1^{-1} = g_1 \Bigg( \Bigg( \sum_{j=2}^L \left( \mathcal{O}_j^{(R)} \cdot \mathcal{O}_{j+1}^{(\overline{R})} + \mathcal{O}_j^{(\overline{R})} \cdot \mathcal{O}_{j+1}^{(R)} \right) \Bigg) \\
+ \mathcal{O}_1^{(R)} \cdot g \left( \mathcal{O}_2^{(\overline{R})} \right) + \mathcal{O}_1^{(\overline{R})} \cdot g \left( \mathcal{O}_2^{(R)} \right) \Bigg) g_1^{-1} \\
= \sum_{j=2}^{L-1} \left( \mathcal{O}_j^{(R)} \cdot \mathcal{O}_{j+1}^{(\overline{R})} + \mathcal{O}_j^{(\overline{R})} \cdot \mathcal{O}_{j+1}^{(R)} \right) \\
+ \left( \mathcal{O}_L^{(R)} \cdot g \left( \mathcal{O}_1^{(\overline{R})} \right) + \mathcal{O}_L^{(\overline{R})} \cdot g \left( \mathcal{O}_1^{(R)} \right) \right) \\
+ \left( g \left( \mathcal{O}_1^{(R)} \right) \cdot g \left( \mathcal{O}_2^{(\overline{R})} \right) + g \left( \mathcal{O}_1^{(\overline{R})} \right) \cdot g \left( \mathcal{O}_2^{(R)} \right) \right) \\
= \lambda^{-1} (W + \Delta W_g^1).
\end{aligned}
\tag{30}
$$

We used the fact that,[16]

$$g \left( \mathcal{O}_j^{(R)} \right) \cdot g \left( \mathcal{O}_{j+1}^{(\overline{R})} \right) = g \left( \mathcal{O}_j^{(R)} \cdot \mathcal{O}_{j+1}^{(\overline{R})} \right) = \mathcal{O}_j^{(R)} \cdot \mathcal{O}_{j+1}^{(\overline{R})}. \tag{31}$$

---

[15]Finally, let us count all the deformations of the theory, whether they preserve or break sub-groups of O(2) symmetry. There are $4 \times L$ nearest neighbour deformations, some of which break the U(1) symmetry and some the $\mathbb{Z}_2$ symmetry. Thus in total we have $4 \times L - L = 3 \times L$ nearest neighbour exactly marginal deformations.

[16]By definition of the product $\cdot$ we project on a $G$ singlet.

Thus we deduce that we have an extension of the $\mathbb{Z}_L$ symmetry preserved,

$$(g_1 T)^L = \prod_{i=1}^{L} g_i = g\,. \tag{32}$$

We can perform this transformation for every lattice link and as before derive that deformations at different links are equivalent. If we focus only on the connected components of a Lie group $G$ we generate $\dim G$ dimensional conformal manifold in this way. We can repeat the construction as before by placing the deformations at different sites, moving, and merging them.

## 4.2 Constructions with gauging symmetries

Let us consider a situation which does not satisfy the conditions of the previous subsection. Again, we discuss an example which can be generalized in multiple ways in a straightforward manner.

Take the theory $\mathcal{T}$ to have operators $\mathcal{O}$ of $R$-charge one in representation $R_+$ of $G = U(1) \times H$ ($H$ semi-simple, representation $R$ of $H$ and charge $+1$ of U(1)). We cannot couple different copies of it in an exactly marginal manner by a superpotential as the relevant Kähler quotient is empty. In such cases however, in $D = 4$ we might be able to come up with an exactly marginal deformation by gauging some subgroup of $H$. This happens if we can find a semi-simple subgroup $H'$ of $H$ such that the 't Hooft anomaly $\operatorname{Tr} R H'^2$ is equal to $-\frac{1}{2}\dim H'$ and then gauging diagonal combination of $H'$ for two copies of $\mathcal{T}$ is marginal. In addition to gauging $H'$ turning on the superpotential,

$$W = \lambda\, \mathcal{O}_1 \cdot \mathcal{O}_2\,, \tag{33}$$

with $\lambda$ proportional to the gauge coupling will produce an exactly marginal deformation. This procedure preserves a diagonal combination of the normalizer of $H'$ in $H$ as well as a diagonal combination of the two U(1)s with the other combination broken by an anomaly due to gauging. The analysis of conformality in this case is the same as the one performed in previous sections.

The lattice constructions discussed here can be generalized in various ways. For example we can consider lattices of higher dimensionality. Moreover we can consider gluings of different theories by multiplying operators $R$-charges of which sum up to 2, or even simultaneous gluings of more than two theories. We proceed with the simple one dimensional lattice examples and discuss higher-dimensional lattices later on.

# 5 Simple examples

## 5.1 $\mathcal{T}$ : SU(2) with $N_f = 4$

Let us now give a concrete example of $\mathcal{T}$ with marginal operators of $R$-charge one charged under O(2) symmetry. We start with probably the simplest interacting supersymmetric QFT in four dimensions, SQCD with gauge group SU(2) and $N_f = 4$: the theory has an octet of chiral superfields in the fundamental representation of SU(2) which we will denote by $Q_i$. This theory has an SU(8) global symmetry. The superconformal $R$-symmetry of the matter fields is $\frac{1}{2}$. We have gauge invariant operators with $R$-charge one, $\mathcal{O}_{ij} = Q_i Q_j$, forming irrep **28** of SU(8). The conformal manifold of this theory was analyzed in detail in [40] and here we will seek a particular direction on it preserving an O(2) subgroup of SU(8). As was argued in [40] one can build marginal operators of the form $\mathcal{O}_{ij}\mathcal{O}_{kl}$, not all of which are independent.

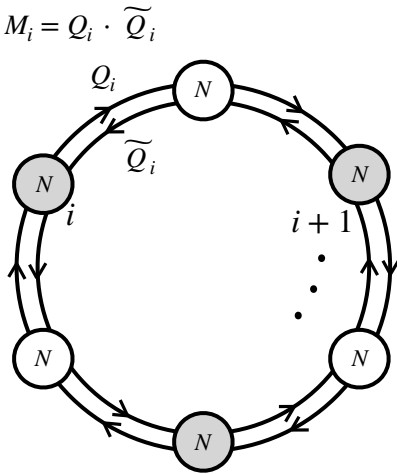

Figure 2: One dimensional lattice with gauging.

In fact, though naively one would expect $\frac{28 \times 29}{2} = 406$ such operators, 70 are missing due to trace relations, and the marginal operators form irrep **336** of SU(8). Some of these marginal operators are exactly marginal. Utilizing these exactly marginal deformations we can build a theory with O(2) symmetry: the construction is detailed in Appendix A. Then we will be exactly in the setup we have discussed till now.

We can generalize this setup a bit. Let us consider the theory without any superpotential which will preserve the full SU(8) symmetry. In fact in [40] pairs of such theories were glued together with the goal to show that the combined theory has a locus on its conformal manifold with the SU(8) symmetry enhancing to $E_7$.[17] Let us consider coupling the theories preserving SU(8) symmetry,

$$W = \lambda \sum_{j=1}^{L} (Q_m^j Q_n^j)(Q_m^{j+1} Q_n^{j+1}). \tag{34}$$

In this case we can repeat our analysis with the non-abelian group $G = $ SU(8). One can construct exactly marginal deformations in the adjoint representation of SU(8) which are topological on the lattice.[18]

## 5.2 $\mathcal{T}$ : SU($N$) with $N_f = 2N$ with gauging

Let us now consider taking $s$ copies of a theory we will denote $\mathcal{T}_0$, SU($N$) with $N_f = 2N$, and gluing them together in the following way. We decompose the SU($2N$) × SU($2N$) × U(1)$_t$ symmetry to SU($N$)$_+$ × U(1)$_+$ × SU($N$)$_-$ × $U(1)_-$ × U(1)$_t$, so that each SU($N$) has TrSU($N$)$_\pm^2 R = -\frac{N}{2}$. We have operators $M^\pm$ in the adjoint irrep of SU($N$)$_\pm$, having $R$-charge one, and neutral under all U(1)$_\pm$ symmetries. These operators are taken to be charged $\pm 1$ under the additional U(1)$_t$ symmetry. Gluing two theories together we gauge a diagonal combination of SU($N$)$_+$ of one

---

[17]It was later argued that in fact gluing pairs of such theories when SU(8) enhances to $E_7$ an additional U(1) emerges [41].

[18]Let us mention here that the lattice theory preserving the SU(8) symmetry can be obtained as relevant deformation of the $D = 4$ model one obtains by compactifying the rank one E-string theory on a torus with flux preserving $E_7$ × U(1) subgroup of the $E_8$ global symmetry in $D = 6$ [20]. Before the deformation we have a circular quiver of $SU(2)$ gauge theories with $N_f = 6$. We again can think of the quiver as a lattice with the number of sites related to the flux. We note that there the number of exactly marginal deformations is exactly 8 for large enough value of the flux. This happens because the relevant Kähler quotient is not trivial only when we close the quiver into a ring.

theory with $\mathrm{SU}(N)_-$ of the other one and turn on the superpotential $M_+ \cdot M'$. Both of these deformations are marginal and turning them on together is an exactly marginal deformation. Gluing $s$ copies we thus have the superpotential,

$$W = \lambda \sum_{i=1}^{s} \left( M_i^+ \cdot M_{i+1}^- + c\, \mathrm{Tr} W_\alpha^{(i)} W^{(i)\alpha} \right), \tag{35}$$

where $c$ is an unimportant numerical coefficient and $W_\alpha^{(i)}$ is the vector field strength chiral superfield of gluing the $i$th copy with the $(i+1)$st. See Figure 2. After gluing the different copies the diagonal combination of all the $\mathrm{U}(1)_t$ symmetries remains as the continuous global symmetry.[19] Before gluing, the different copies of $\mathcal{T}_0$ have their own $\mathrm{U}(1)_t$ symmetry. The non conservation equation takes the following form,

$$\overline{\mathrm{D}}^2 J_i(\mathbf{x}) = X(\lambda) \left( M_i^+ \cdot M_{i+1}^- - M_{i-1}^+ \cdot M_i^- - c\, \mathrm{Tr} W_\alpha^{(i)} W^{(i)\alpha} + c\, \mathrm{Tr} W_\alpha^{(i-1)} W^{(i-1)\alpha} \right). \tag{36}$$

We have the contribution of the vectors as the U(1) symmetry is anomalous. Thus we can define

$$\mathcal{J}(\mathbf{x}, i) = X(\lambda) \left( c\, \mathrm{Tr} W_\alpha^{(i)} W^{(i)\alpha} - M_i^+ \cdot M_{i+1}^- \right), \tag{37}$$

so that

$$\overline{\mathrm{D}}^2 J_i(\mathbf{x}) + \mathcal{J}(\mathbf{x}, i) - \mathcal{J}(\mathbf{x}, i-1) = 0. \tag{38}$$

We thus see that we have an exactly marginal deformation $\Delta W = \mathcal{J}(\mathbf{x}, i)$ which is topological on the lattice in the sense we have discussed before.[20]

# 6 $D = 2$ lattices

Let us consider a generalization of the discussion to two dimensional square lattices. Most of the considerations can be repeated verbatim and here we stress some of the new features. Again, we do so by discussing an example. We consider coupling $L_1 \times L_2$ copies of $\mathcal{T}$ with a superpotential,

$$W = \lambda \sum_{i=1}^{L_1} \sum_{j=1}^{L_2} \left( \mathcal{O}_{i,j}^+ \mathcal{O}_{i,j+1}^- + \mathcal{O}_{i,j}^- \mathcal{O}_{i,j+1}^+ + \mathcal{O}_{i,j}^+ \mathcal{O}_{i+1,j}^- + \mathcal{O}_{i,j}^- \mathcal{O}_{i+1,j}^+ \right), \tag{39}$$

with $\mathcal{O}_{L_1+1,j}^\pm = \mathcal{O}_{1,j}^\pm$ and $\mathcal{O}_{i,L_2+1}^\pm = \mathcal{O}_{i,1}^\pm$. We thus have a toroidal lattice with periodic boundary conditions on the two cycles. This coupling is exactly marginal and preserves an O(2) diagonal subgroup of the global symmetry.

The non-conservation equation at each lattice site takes the form

$$\begin{aligned}
\overline{\mathrm{D}}^2 J_{i,j}(\mathbf{x}) = X(\lambda) \Big[ &\left( \mathcal{O}_{i,j}^+ \mathcal{O}_{i+1,j}^- - \mathcal{O}_{i,j}^- \mathcal{O}_{i+1,j}^+ \right) - \left( \mathcal{O}_{i-1,j}^+ \mathcal{O}_{i,j}^- - \mathcal{O}_{i-1,j}^- \mathcal{O}_{i,j}^+ \right) \\
&+ \left( \mathcal{O}_{i,j}^+ \mathcal{O}_{i,j+1}^- - \mathcal{O}_{i,j}^- \mathcal{O}_{i,j+1}^+ \right) - \left( \mathcal{O}_{i,j-1}^+ \mathcal{O}_{i,j}^- - \mathcal{O}_{i,j-1}^- \mathcal{O}_{i,j}^+ \right) \Big].
\end{aligned} \tag{40}$$

---

[19] We also have $2s$ copies of U(1) symmetries coming from $\mathrm{U}(1)_\pm$ from each $\mathcal{T}_0$. These symmetries are localized on the lattice and are examples of sub-system symmetries. The lattice translations act on these symmetry. Thus the $\mathrm{U}(1)^{2s}$ symmetry is extended by the lattice translations.

[20] Note that we can repeat the analysis with some obvious modifications when all the $\mathrm{SU}(N)$ involves $\mathcal{N} = 2$ vector multiplets and superpotentials. In this case we will obtain $2s$ $\mathcal{N} = 2$ preserving deformations and an additional $\mathcal{N} = 1$ one (See e.g. [42].). The latter is the topological deformation.

We define

$$\mathcal{J}_{(D+1)}(\mathbf{x},i,j) \equiv X(\lambda)\left(\mathcal{O}_{i-1,j}^{-}\mathcal{O}_{i,j}^{+} - \mathcal{O}_{i-1,j}^{+}\mathcal{O}_{i,j}^{-}\right),$$
$$\mathcal{J}_{(D+2)}(\mathbf{x},i,j) \equiv X(\lambda)\left(\mathcal{O}_{i,j-1}^{-}\mathcal{O}_{i,j}^{+} - \mathcal{O}_{i,j-1}^{+}\mathcal{O}_{i,j}^{-}\right),$$
(41)

and then the non conservation can be written as

$$\overline{\mathrm{D}}^2 J_{i,j}(\mathbf{x}) + \mathcal{J}_{(D+1)}(\mathbf{x},i+1,j) - \mathcal{J}_{(D+1)}(\mathbf{x},i,j) + \mathcal{J}_{(D+2)}(\mathbf{x},i,j+1) - \mathcal{J}_{(D+2)}(\mathbf{x},i,j) = 0, \quad (42)$$

which has the structure of a conservation equation in $D + 2$ dimensions with the extra two dimensions discretized. In particular, we can write

$$\partial_\mu j^\mu(\mathbf{x},i) + j^{(D+1)}(\mathbf{x},i+1,j) - j^{(D+1)}(\mathbf{x},i,j) + j^{(D+2)}(\mathbf{x},i,j+1) - j^{(D+2)}(\mathbf{x},i,j) = 0, \quad (43)$$

where we have taken the F-term of (11) and defined,

$$j^{(D+\ell)}(\mathbf{x},i) = \int d^2\theta \, \mathcal{J}_{(D+\ell)}(\mathbf{x},i) - \text{c.c.} \quad (44)$$

We can now construct two exactly marginal deformations, corresponding to the two cycles of the torus, which are topological on the torus,

$$\mathcal{O}_{(D+1)}(i) = \sum_{j=1}^{L_2} \int d^4x \int d^2\theta \, \mathcal{J}_{(D+1)}(\mathbf{x},i,j),$$
$$\mathcal{O}_{(D+2)}(j) = \sum_{i=1}^{L_1} \int d^4x \int d^2\theta \, \mathcal{J}_{(D+2)}(\mathbf{x},i,j).$$
(45)

The deformations satisfy that

$$\mathcal{O}_{(D+1)}(i) \sim \mathcal{O}_{(D+1)}(i+1), \qquad \mathcal{O}_{(D+2)}(j) \sim \mathcal{O}_{(D+2)}(j+1). \quad (46)$$

These deformations can be thought of, as before, as *defects* for the U(1) symmetry. Note that in fact we have more topological deformations summing over any curve on the dual lattice. However, if the curve has zero winding number then the corresponding marginal deformation is equivalent to no deformation: that is it is a marginally irrelevant deformation.

In addition to the topological marginal operators we also have non-topological nearest neighbour deformations. A class of such deformations is given by introducing different couplings for every edge of the lattice,

$$\Delta W_{ij}^\uparrow \equiv \mathcal{O}_{i,j}^+\mathcal{O}_{i,j+1}^- + \mathcal{O}_{i,j}^-\mathcal{O}_{i,j+1}^+, \qquad \Delta W_{ij}^\rightarrow \equiv \mathcal{O}_{i,j}^+\mathcal{O}_{i+1,j}^- + \mathcal{O}_{i,j}^-\mathcal{O}_{i+1,j}^+. \quad (47)$$

We have $2 \times L_1 \times L_2$ exactly marginal deformations of this sort corresponding to the edges of the lattice. We have an additional set of marginal deformations which we can associate to the lattice faces,

$$\Delta W_{ij}^\circ \equiv \mathcal{J}_{(D+1)}(\mathbf{x},i,j) + \mathcal{J}_{(D+2)}(\mathbf{x},i+1,j) - \mathcal{J}_{(D+1)}(\mathbf{x},i,j+1) - \mathcal{J}_{(D+2)}(\mathbf{x},i,j). \quad (48)$$

Note that these are exactly marginal and independent of the deformations corresponding to the edges. We have $L_1 \times L_2 - 1$ such deformations. The $-1$ comes from the fact that turning on all such deformations but one is equivalent to turning on the remaining deformation.

Thus in total we have $2 \times L_1 \times L_2$ exactly marginal deformations corresponding to edges of the lattice. We have $L_1 \times L_2 - 1$ independent exactly marginal deformations corresponding to faces of the lattice. Finally, there are two exactly marginal deformations which are associated to the cycles of the torus defining the lattice. The latter deformations are topological on the

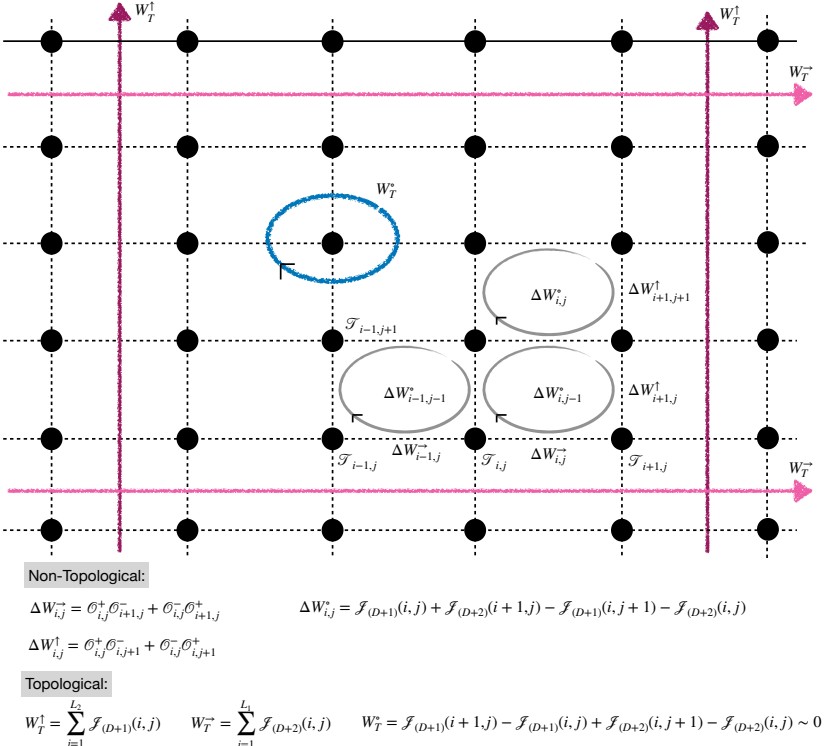

Figure 3: The two dimensional lattice and the various types of exactly marginal deformations. We have topological deformations on the lattice corresponding to the cycles of the torus which are non-trivial; topological and trivial (marginally irrelevant) operators corresponding to paths on the dual lattice which do not wind the cycles; and non topological exactly marginal deformations.

lattice while the former are not topological. Moreover, the lattice has two translation symmetries around the two cycles, and thus even in presence of the two topological deformations we can preserve a twisted version of these lattice translations.[21]

# 7 Compactifications of $D = 6$ SCFTs to $D = 4$

We discuss a simple concrete example of a $D = 2$ lattice with superpotential interactions in Appendix B. Here we will consider a broad class of more involved examples. As a class of natural examples of two dimensional lattices we will consider compactification of a generic $D = 6$ SCFT on a Riemann surface of genus $g = L_1 \times L_2 + 1$. Such a compactification can be described in $D = 4$ as gluing of compactifications on $L_1 \times L_2$ four punctured spheres.

We start the construction by choosing a $D = 6$ $(1,0)$ SCFT (see [4] for a review) and consider a compactification of it down to $D = 4$ on a closed Riemann surface. If the theory in $D = 6$ has a continuous global symmetry $\mathfrak{g}_{6d}$ then one can turn on flux for abelian subgroups of $\mathfrak{g}_{6d}$ supported on the Riemann surface.[22] Examples of $D = 6$ $(1,0)$ SCFTs include among

---

[21]The total number of marginal operators that we consider is two operators per every edge, *e.g.* $\mathcal{O}_{i,j}^+ \mathcal{O}_{i+1,j}^-$ and $\mathcal{O}_{i,j}^- \mathcal{O}_{i+1,j}^+$, and thus $4 \times L_1 \times L_2$ operators. The number of symmetries before turning on interactions is one per site, $L_1 \times L_2$, and after turning on interactions one symmetry remains. Thus, the number of exactly marginal operators is expected to be $4 \times L_1 \times L_2 - L_1 \times L_2 + 1$. This is what we obtain, $2 \times L_1 \times L_2 + (L_1 \times L_2 - 1) + 2$.

[22]For the discussion here the global structure of the symmetry group is immaterial and thus in fact we will discuss the symmetry groups in terms of their algebra.

others: the $(2,0)$ theories with $\mathfrak{g}_{6d} = \mathfrak{su}(2)$; the rank one E-string theory with $\mathfrak{g}_{6d} = \mathfrak{e}_8$; $(D,D)$ minimal conformal matter theories with $\mathfrak{g}_{6d} = \mathfrak{so}(4N+12)$; the minimal $D=6$ SCFTs with $\mathfrak{g}_{6d} = \emptyset$. We will set the flux for abelian sub-groups of $\mathfrak{g}_{6d}$ to zero. The compactification surface is a smooth geometry and does not have a natural sub-structure for generic values of complex structure parameters: one can decompose it for example into gluings of three punctured spheres (pairs-of-pants) but there is no preferred way to do so. However, in particular limits of the complex structure moduli one of the decompositions becomes more natural than others. Here we will consider a very specific decomposition and argue that it provides a surprising new perspective on such compactifications. We will organize the theory as a planar square lattice by decomposing it into four-punctured sphere building blocks. Taking the genus to be $g = L_1 \times L_2 + 1$ $(L_i \in \mathbb{N})$ we will have a square periodic lattice with $L_1 \times L_2$ vertices/sites.

We assume that compactifying the $D=6$ SCFT on a circle, possibly with holonomies for $\mathfrak{g}_{6d}$, we obtain an effective SQFT in $D=5$ given in terms of an $\mathcal{N}=1$ gauge theory with gauge group $\mathfrak{g}_{5d}$. Examples include among others: for the $(2,0)$ ADE theories $\mathfrak{g}_{5d} = \mathfrak{ade}$; for the rank $k$ E-string theory $\mathfrak{g}_{5d} = \mathfrak{usp}(2k)$ or $\mathfrak{su}(k+1)_{\pm\frac{k+1}{2}}$; for $(D,D)$ minimal conformal matter theories we have a choice of $\mathfrak{g}_{5d} = \mathfrak{su}(N+1)$, $\mathfrak{usp}(2N)$, $\mathfrak{su}(2)^N$; for the minimal $A_2$ $D=6$ SCFT $\mathfrak{g}_{5d} = \mathfrak{su}(3)$. See *e.g.* [43–46]. Such gauge theory descriptions in $D=5$ can be used to define compactifications on *punctured* Riemann surfaces by analyzing possible supersymmetric boundary conditions for the $D=5$ fields at the punctures. In particular one can define *maximal punctures* such that each puncture comes with a copy of global symmetry equal to $\mathfrak{g}_{5d}$. See [32] for a review.

The basic building block of our construction is compactification on a sphere with four maximal punctures and zero flux. The four punctured sphere ($4pt$ sphere) has (for a generic $D=6$ theory) a single exactly marginal operator in $D=4$ corresponding to the complex structure modulus. The boundary conditions at the punctures break the $\mathfrak{g}_{6d}$ symmetry to a subgroup. We will denote the group which is preserved as $\mathfrak{g}_P \subseteq \mathfrak{g}_{6d}$. The symmetry $\mathfrak{g}_P$ depends on the type of the puncture. The type of the puncture is encoded in the set of *moment map operators*: these are operators of charge $+1$ under the Cartan of the $D=6$ $\mathfrak{su}(2)$ R-symmetry (the fundamental of $\mathfrak{su}(2)$ has charges $\pm\frac{1}{2}$), and which are charged under the puncture symmetry. We will denote the moment map operators by $M_P$. We will assume that the symmetry $\mathfrak{g}_P$ is either empty (see *e.g* [47]) or it has a $\mathfrak{u}(1)_t$ factor under which all the components of the moment map have the same charge. In the latter case we will denote $\mathfrak{g}_P = \mathfrak{u}(1)_t \times \tilde{\mathfrak{g}}_P$. The representation $\rho_P$ of $M_P$ is such that decomposing $\rho_P \times \bar{\rho}_{P'}$ into irreps of $\mathfrak{g}_{5d}^{(diag.)} \times \mathfrak{g}_P \times \mathfrak{g}_{P'}$, the singlet of $\mathfrak{g}_{5d}^{(diag.)}$ multiplies only the bi-fundamental irrep of $\mathfrak{g}_P \times \mathfrak{g}_{P'}$.[23]

Different types of punctures have $M_P$ in different representations of $\mathfrak{g}_{6d}$. We will take the punctures of our chosen $4pt$ sphere to come in two identical conjugate pairs. This together with the fact that the flux is zero guarantees that the Cartan generator of the $D=6$ $\mathfrak{su}(2)$ R-symmetry is the superconformal R-symmetry of the $D=4$ $\mathcal{N}=1$ theory (as usual, barring accidental appearance of $\mathfrak{u}(1)$ symmetries in $D=4$). The theory also has marginal operators. There are marginal operators which are charged under puncture symmetries and such which are not. The latter come in a representation of $\mathfrak{g}_P$ which compliments the adjoint of $\mathfrak{g}_P$ to the adjoint of $\mathfrak{g}_{6d}$. In particular the character of the marginal operators is [19],

$$\chi_{adj.\,\mathfrak{g}_{6d}}(\mathbf{u}) - \chi_{adj.\,\mathfrak{g}_P}(\mathbf{u}). \tag{49}$$

Note that this theory has an S-duality group acting on the one dimensional conformal manifold associated to the complex structure modulus. This duality is related to different pair of pants decompositions [9]. We tune the exactly marginal coupling to a special locus to be discussed soon. See Appendix C for an explicit example of a $4pt$ sphere.

---

[23]These assumptions are realized in many worked out examples. See [48–52].

Next we consider gluing the $4pt$ sphere building blocks into a closed surface. To glue two conjugate punctures, $P_1$ and $P_2$, we gauge a diagonal combination of the two puncture $\mathfrak{g}_{5d}$ symmetries and turn on a superpotential coupling the two moment map operators,

$$W = M_{P_1} \cdot M_{P_2}. \tag{50}$$

The gauging here is marginal as the superconformal R-symmetry is the Cartan of the $D = 6$ R-symmetry and with it $\mathrm{Tr}\, R\, \mathfrak{g}_{5d}^2 = 0$ [32]. The superpotential is marginal as the R-charges of the moment maps are +1. Turning on both, the superpotential and the gauging, gives an exactly marginal deformation. The operator $M_{P_1} \cdot M_{P_2}$ is in representation $\rho_{P_1} \times \bar{\rho}_{P_2}$. When we turn this deformation on we project this on a singlet of $\mathfrak{g}_{5d}$. The $\mathfrak{g}_{5d}$ singlet part of $M_{P_1} \cdot M_{P_2}$ is in bi-fundamental irrep of $\mathfrak{g}_{P_1} \times \mathfrak{g}_{P_2}$ by assumption. We can always construct a singlet of $\mathfrak{g}_{P_1} \times \mathfrak{g}_{P_2}$ from such a bi-fundamental representation. Thus the Kähler quotient built from marginal operators is not empty [13] implying that the exactly marginal deformation preserves a diagonal combination of the two $\mathfrak{g}_{P_i}$ symmetries. After identification of the two $\mathfrak{g}_{P_i}$ the adjoint irrep of $\mathfrak{g}_P$ in $M_{P_1} \cdot M_{P_2}$ recombines with the broken part of the conserved currents of the two $\mathfrak{g}_{P_i}$. The marginal operators of the combined theory (which are singlets of the puncture symmetry) form two copies of (49).

We can repeat this procedure to form a sphere with $s$ punctures. Since we glue $4pt$ spheres, $s$ is even. The symmetry is $\mathfrak{g}_P$ and the number of exactly marginal operators preserving it is $s-3$. We have additional $\frac{s-2}{2}$ marginal operators in representation (49) of $\mathfrak{g}_P$. Finally we can glue the punctures in conjugate pairs to form a higher genus surface. All of these gaugings are exactly marginal and the superpotentials (50) are exactly marginal by themselves as these do not break any symmetry when conjugate punctures of the same surface are glued. Each such gluing produces 2 exactly marginal deformations preserving $\mathfrak{g}_P$ along with a marginal operator in adjoint representation of $\tilde{\mathfrak{g}}_P$. If we glue all $s$ punctures of the sphere in pairs to form a genus $g = \frac{s}{2}$ surface, the number of exactly marginal operators is then $(s-3) + s = 3g - 3 + g$. We interpret the $3g-3$ deformations as corresponding to the complex structure moduli while the additional $g$ deformations correspond to flat connection for the $\mathfrak{u}(1)_t$ [19, 42].[24] We have the following marginal operators not corresponding to complex structure moduli,

$$\frac{s-2}{2} \left( \chi_{adj.\mathfrak{g}_{6d}}(\mathbf{u}) - \chi_{adj.\mathfrak{g}_P}(\mathbf{u}) \right) + \frac{s}{2} \chi_{adj.\mathfrak{g}_P}(\mathbf{u}) = (g-1) \chi_{adj.\mathfrak{g}_{6d}}(\mathbf{u}) + \chi_{adj.\mathfrak{g}_P}(\mathbf{u}). \tag{51}$$

These will play a role in what follows.[25]

## 7.1 The lattice and the symmetries

We interpret compactification on genus $g$ surface as a lattice construction as follows. Starting from $L_1 \times L_2$ $4pt$ spheres one builds a genus $g = L_1 \times L_2 + 1$ surface. We glue so that an $L_1 \times L_2$ square doubly periodic lattice is produced. See Fig. 4. We can associate the exactly marginal deformations coming from complex structure moduli to the lattice: the $L_1 \times L_2$ deformations coming from the lattice sites ($4pt$ spheres) and $2L_1 \times L_2$ associated with the edges of the lattice. One can think of the couplings for the latter as setting the lengths of the edges. The superpotential for the genus $g$ model can be written as

$$W = \sum_{i=1}^{L_1} \sum_{j=1}^{L_2} \left( \lambda_{i,j}^- M_{i,j}^{\rightarrow} \cdot M_{i+1,j}^{\leftarrow} + \lambda_{i,j}^| M_{i,j}^{\downarrow} \cdot M_{i,j+1}^{\uparrow} \right). \tag{52}$$

---

[24]Note that if $\mathfrak{g}_P$ is empty we do not have moment maps and we do not have exactly marginal operators in addition to complex structure moduli.

[25]Note that the exactly marginal deformations corresponding to gluing the $4pt$ spheres together (the superpotential and gauging tuned to be exactly marginal) correspond to some combination of the flat connection for $\mathfrak{u}(1)_t$ and the complex structure modulus. We can tune this combination so that the holonomy part will be zero. On that strongly coupled locus of the conformal manifold the symmetry $\mathfrak{g}_P$ enhances to $\mathfrak{g}_{6d}$.

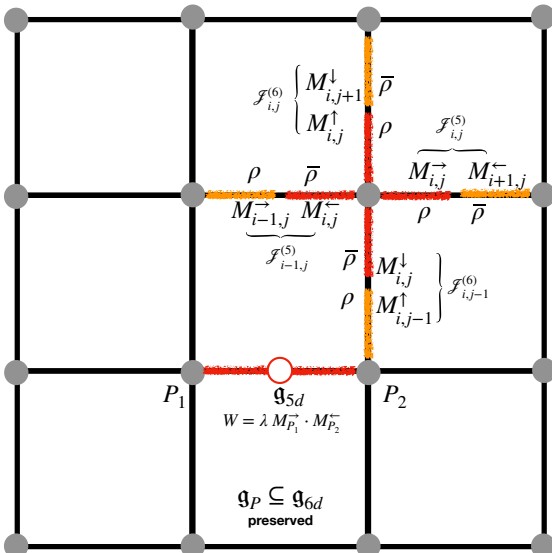

Figure 4: The square lattice.

This superpotential preserves $\mathfrak{g}_P$. Note that the superpotential also includes terms due to gauging which we omit for the sake of brevity. We label the moment maps with lattice position and arrows associated to the orientation of the corresponding punctures. Note that in principle by splitting the $4pt$ spheres into three punctured spheres one can obtain a lattice with trivalent vertices. However, generally gluing three punctured spheres into $4pt$ sphere is not done by exactly marginal coupling but by a deformation which involves an RG flow.

Let us consider turning on the above superpotential with equal small exactly marginal couplings $\lambda = \lambda^-_{i,j} = \lambda^|_{i,j}$. In this case the theory is invariant under the lattice translations and we are exactly in the general setup that we have discussed in the previous sections. With $\lambda = 0$ we have a copy of $\mathfrak{g}_P$ for each lattice site. Turning on the interactions this symmetry is broken to the diagonal one. As before, we can still define operators $J_{ij}(\mathbf{x})$ which are the (linear supermultiplet of) conserved currents if $\lambda = 0$. We also have (chiral) marginal operators $\mathcal{J}^{(5)}_{ij}(\mathbf{x}) \equiv M^\rightarrow_{i,j} M^\leftarrow_{i+1,j}$ and $\mathcal{J}^{(6)}_{ij}(\mathbf{x}) \equiv M^\downarrow_{i,j} M^\uparrow_{i,j+1}$. These operators are in the representation $\rho_P \times \bar\rho_P$ of two copies of $\mathfrak{g}_P$. As we discussed before [13] the non conservation of the currents $J_{ij}(\mathbf{x})$ is related to the superpotential we turn on,

$$\overline{\mathrm{D}}^2 J^a_{ij}(\mathbf{x}) = X(\lambda)\, T^{(\rho)a} \left( \mathcal{J}^{(5)}_{ij}(\mathbf{x}) - \mathcal{J}^{(5)}_{(i-1)j}(\mathbf{x}) + \mathcal{J}^{(6)}_{ij}(\mathbf{x}) - \mathcal{J}^{(6)}_{i(j-1)}(\mathbf{x}) \right). \tag{53}$$

Here $a$ labels a generator of $\mathfrak{g}_P$ and $T^{(\rho)a}$ is the representation of that generator in $\rho$. Strictly speaking the above equation is correct for the $\tilde{\mathfrak{g}}_P$ part of the $\mathfrak{g}_P = \mathfrak{u}(1)_t \times \tilde{\mathfrak{g}}_P$. The $\mathfrak{u}(1)_t$ non-conservation equation will include also the contribution from the gauging and we do not write it explicitly not to clatter notations. The relative sign between the different terms is due to the conjugation. The divergence of the current appears in the (imaginary part of) the F-term of the chiral fields above. This non conservation, as in our general discussion, has a structure of,

$$\partial_\mu j^\mu_a(\mathbf{x}, i, j) + \Delta_5\, j^{(5)}_a(\mathbf{x}, i, j) + \Delta_6\, j^{(6)}_a(\mathbf{x}, i, j) = 0, \tag{54}$$

where $j^\mu_a$ is the current at site $(i, j)$ and $\Delta_I$ is a discretized derivative. *Thus the non-conservation of the currents at each lattice point has a structure of a conservation law once we include two additional discretized lattice directions.* We can construct topological charge operator, as was done before, which will give rise to exactly marginal deformations corresponding to the two

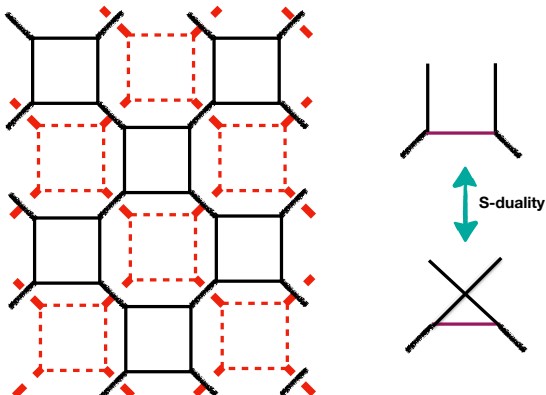

Figure 5: Thinking about the lattice as constructed from three punctured spheres we have a lattice built from squares and octagons. The thicker lines connect three punctured spheres into the $4pt$ sphere. The duality of all the edges gluing three punctured spheres into four punctured ones exchanges the octagons with squares. At the self dual point of the $4pt$ spheres we consider these two lattices are equivalent. The duality exchanges either the vertical lines or horizontal ones. Moreover, the edges connecting different $4pt$ spheres are tuned to be at a self dual point as described on the RHS of the figure: the relevant duality exchanges the lines corresponding to lattice edges. This is the self dual locus we consider in the paper.

cycles of the lattice. Note that we can add plaquette deformations of the form (48). We will have $L_1 \times L_2 - 1$ independent deformations of this sort.[26]

## 7.2 Duality and the continuum limit

As mentioned above, some of the couplings related to complex structure moduli can be viewed as roughly the lengths of different edges on the lattice. Taking these couplings to be weak we are going to the limit of *long edge sizes*. Increasing the magnitude of the couplings decreases the length of the edges. However, increasing too much we move to a different weakly coupled *duality* frame increasing the length of the edge again. The dualities effectively rewire the lattice.[27] In this sense a minimal length is achieved at the self-dual point at which the model is "maximally" strongly coupled. To be more precise about the duality it is beneficial to think about the surface as constructed from three punctured spheres, see Fig. 5. We then first tune couplings of each $4pt$ sphere to a self dual locus as described in the Figure.[28] We then also consider the duality exchanging the edges which are connecting different $4pt$ spheres. *A natural notion of going to zero lattice spacing is thus tuning all the couplings to be at the self-dual point.* Keeping the genus, that is the size of the system, finite the theory becomes strongly coupled and we loose all notion of locality on the lattice. However, we will want to take a double scaling limit of going to the self-dual point and taking the genus to infinity ($L_i \to \infty$), in a correlated way. This parallels the continuum limit in the Heisenberg model where one

---

[26]We can think of the plaquette deformations as turning on a background gauge field for the symmetry with components along the lattice directions non vanishing and independent of the continuous directions. Moreover, the exactly marginal deformations of the $D = 4$ theory are coming from holonomies for $\mathfrak{g}_{6d}$ on the compactification surface [18, 19, 42]. For non-abelian symmetries these deformations break the symmetries and thus their number is smaller by 1 per current component.

[27]Let us stress that as we preserve here conformality the dualities we have in mind are *conformal* and not IR (Seiberg) dualities.

[28]Note that the compactifications that we are discussing sometimes can be realized as brane webs, see *e.g.* [53]. The duality transformations then have interpretations as certain manipulations of such brane webs.

takes the lattice spacing to zero and the number of sites to infinity. We thus can conjecture that in the double scaling limit discussed above the two dimensional lattice of the definition of the theory becomes a continues two dimensional space. Since the lattice is constructed from copies of identical $D = 4$ theories, we have a natural notion of local operators on the lattice $\mathcal{O}_{i,j}(\mathbf{x})$. For example, some of these are gauge invariant operators of the $4pt$ sphere at lattice point $(i, j)$. To retain some notion of locality when considering correlation function of such operators we will scale their lattice distance also to infinity. For example consider the two point function, $\langle \mathcal{O}_{P_1}(\mathbf{x}) \mathcal{O}_{P_2}(\mathbf{0}) \rangle$, where $P_1$ and $P_2$ label lattice points. For weak couplings this expectation value decreases as we increase the lattice distance between $P_1$ and $P_2$. On the other hand for the self-dual locus the duality removes the notion of locality on the lattice. However, we can take a limit where as we tune the coupling to the self dual point, we also scale the lattice distance to infinity and thus still expectation values decrease as the distance $\Delta$ increases: *e.g.* $|\lambda - \lambda_*|^\alpha |P_1(\lambda) - P_2(\lambda)| \sim \Delta$ with $\alpha \in \mathbb{R}_+$, $\lambda_*$ being the self-dual locus and $\Delta$ fixed in the limit. Note that tuning to strong coupling locus we expect the theory to have the full $\mathfrak{g}_{6d}$ as its symmetry, although the lattice construction only manifests a subgroup of it.

Let us comment here that whatever a sensible continuum limit is, if it exists at all, it should involve taking the lattice size to infinity and tuning the couplings. Some quantities that we can compute in $D = 4$ do not depend on the couplings and thus for them this limit can be explicitly computed. Let us illustrate this with the supersymmetric index [38, 54, 55]. In the limit of large size of the lattice, which in our discussion here is the limit of large genus of the compactification surface, it has the following form [56, 57],

$$\mathcal{I}^{(Lattice)} \sim \left( \hat{C}_0(p, q, u_{\mathfrak{g}_{6d}}) \right)^{L_1 \times L_2}, \tag{55}$$

where $p$ and $q$ are fugacities coupling to the generators of the superconformal symmetry in the standard manner [58], and $u_{\mathfrak{g}_{6d}}$ are fugacities for the $D = 6$ global symmetry.[29] The equality holds up to orders in expansion in $p$ and $q$. This expression can be interpreted as having a certain contribution from each point of the continuum torus. Alternatively, as the index counts certain local operators in $D = 4$, this expression can be thought of as averaging of local operators on the continuum torus with arbitrary smearing function. This arbitrary non-locality should be contrasted with taking a $D = 6$ SCFT and obtaining $D = 4$ local operators smearing the $D = 6$ ones on the torus only with a particular set of functions to preserve supersymmetry [18].

## 8 Discussion and open questions

We have considered in this paper a general construction of a class of CFTs by coupling conformally building blocks CFTs. Generally, one can then consider the building blocks as vertices of a graph with edges appearing if two building blocks are connected by an exactly marginal deformation. In principle there might be edges of different types corresponding to different types of exactly marginal deformations. In particular we have focused on graphs forming regular lattices. We have shown that certain current non-conservation equations can be interpreted as

---

[29]The full index of $4pt$ sphere with zero flux and two pairs of conjugated punctures is given by [59, 60],

$$\sum_\lambda \hat{C}_\lambda \psi_\lambda(a) \psi_\lambda(b) \widetilde{\psi}_\lambda(c) \widetilde{\psi}_\lambda(d), \tag{56}$$

where the precise meaning of the sum depends on the $D = 6$ SCFT. The functions $\psi_\lambda(a)$ and $\widetilde{\psi}_\lambda(a)$ capture the contributions of the puncture and its conjugate and depend on fugacities for $\mathfrak{g}_{5d}$. A conjectured interpretation of this expression is that the $\lambda = 0$ is capturing the local operators in $D = 4$ coming from local operators in $D = 6$, while rest of $\lambda$ capture non-local operators in $D = 6$ giving rise to local operators in $D = 4$ [56].

current conservation equation when one also includes the lattice directions. Further we have shown that the extra current components associated to the lattice directions are associated to exactly marginal deformations. In particular we have shown that when these deformations are added to the action with coefficients satisfying certain reality conditions they can be interpreted precisely as defects for some of the global symmetry and are topological on the lattice. We have given several simple examples of the construction and also discussed how such lattices naturally appear once one studies compactifications of $D = 6$ SCFTs on Riemann surfaces.

There are many avenues for further research. For example, as we have commented upon in the case of compactifications, it would be very interesting to understand certain limits of the lattices which could be interpreted as continuum limits. A motivation to consider such limits can be to construct higher dimensional CFTs. We have mentioned that the value of the exactly marginal couplings can be naturally thought to define a notion of the distance on the lattice. Moreover, as often SCFTs have non-trivial dualities relating different values of the couplings (see *e.g* [9, 61]) an interesting continuum limit might correspond to tuning the exactly marginal parameters to special values which are not necessarily singular. It is not clear under which conditions higher dimensional continuous space-time symmetries would emerge and understanding such questions will be very beneficial.

There are several questions one can address without taking any limits. For example, as we have symmetries associated to the lattice, such as lattice translations, one can wonder whether there are any mixed anomalies between the different symmetries of the theory.[30] Another question is whether the lattice constructions of SCFTs allow for generalized versions of symmetries, such as non-invertible ones or higher-form symmetries, see *e.g.* [65–74]. For example, non-invertible symmetries are often related to conformal dualities [75,76] that we have mentioned in the discussion of the continuum limit. Moreover, one can discuss also lattices with *irregularities*, *i.e.* defects. For example, one can explore on-site conformal manifolds for certain sites but not others, or change the SCFT for certain site more drastically still coupling it conformally to the neighboring SCFTs. Such defects and their interplays with generalized notions of symmetry where for example recently discussed in [77].

We leave these and other questions for future investigations.

## Acknowledgments

We are grateful to Ibrahima Bah, Cristopher Beem, Zohar Komargodski, Elli Pomoni, and Gabi Zafrir for very insightful discussions and comments.

**Funding information**     This research is supported in part by the Planning and Budgeting committee, by the Israel Science Foundation under grant no. 2159/22, by BSF grant no. 2018204, and by BSF-NSF grant no. 2023769.

## A   Construction of a theory with $G = O(2)$

We consider the SU(2) $N_f = 4$ SQCD and denote the octet of fields in the fundamental representation of SU(2) by $Q_i$. The mesonic superpotential,

$$W = \lambda_0 \sum_{1 \leq i,j \leq 4} \left( Q_i Q_{j+4} \right) \left( Q_{i+4} Q_j \right),$$ (A.1)

---

[30]See *e.g.* [62–64] for possibly relevant discussions.

is an exactly marginal deformation preserving an SU(4) × U(1) subgroup of SU(8) such that $\mathbf{8} \rightarrow \mathbf{4}_+ \oplus \overline{\mathbf{4}}_-$ [40]. From here the marginal operators decompose as,

$$
\begin{aligned}
\mathbf{336} \quad \rightarrow \quad & 2 \times \mathbf{1}_0 + 2 \times \mathbf{15}_0 + \mathbf{84}_0 + \mathbf{20}'_0 + \mathbf{20}'_4 + \mathbf{20}'_{-4} \\
& + \mathbf{6}_2 + \mathbf{6}_{-2} + \mathbf{10}_2 + \overline{\mathbf{10}}_{-2} + \mathbf{64}_2 + \mathbf{64}_{-2} \,.
\end{aligned}
\tag{A.2}
$$

On the other hand the adjoint irrep of SU(8) decomposes as,

$$
\mathbf{63} \rightarrow \mathbf{1}_0 + 2 \times \mathbf{15}_0 + \mathbf{6}_2 + \mathbf{6}_{-2} + \mathbf{10}_2 + \overline{\mathbf{10}}_{-2} \,.
\tag{A.3}
$$

Thus the marginal operators in,

$$
\mathbf{15}_0 + \mathbf{6}_2 + \mathbf{6}_{-2} + \mathbf{10}_2 + \overline{\mathbf{10}}_{-2} \,,
\tag{A.4}
$$

recombine with the corresponding components of the conserved currents and we are left with marginal operators in,

$$
2 \times \mathbf{1}_0 + \mathbf{15}_0 + \mathbf{84}_0 + \mathbf{20}'_0 + \mathbf{20}'_4 + \mathbf{20}'_{-4} + \mathbf{64}_2 + \mathbf{64}_{-2} \,.
\tag{A.5}
$$

The operators in $\mathbf{1}_0$ are exactly marginal one of them is (A.1) preserving SU(4) × U(1). Next, we turn on $\mathbf{15}_0$ which will break SU(4) to its Cartan sub-group. Finally, we can use the $\mathbf{20}'_0$ to break the Cartan sub-group of SU(4) completely. This will be our theory $\mathcal{T}$. The only symmetry that we are left with is the baryonic U(1). In particular note that the procedure produces a theory invariant under charge conjugation of the U(1) and thus we have actually an O(2) symmetry. We have operators of $R$-charge one charged $\pm 1$ under this O(2). As the quarks are in $\mathbf{8}$ we can build $R$-charge one operators with charges,

$$
\mathbf{28} \rightarrow 6 \times \mathbf{1}_{+1} + 6 \times \mathbf{1}_{-1} + 16 \times \mathbf{1}_0 \,.
\tag{A.6}
$$

We can pick one of the $\mathbf{1}_{+1}$ operators and one of the $\mathbf{1}_{-1}$ operators to be our $\mathcal{O}^{\pm}$ operators. As this exercise exemplifies there are many more choices for various groups and transformations.

## B  An example in $D = 3$

Let us discuss here an example which is slightly different than the general setup discussed in the bulk of the paper. Let us take as our basic theory a $D = 3$ SCFT which is the IR fixed point of the $XYZ$ model. Namely, we have three chiral fields $X$, $Y$, and $Z$, coupled through superpotential,

$$
W = XYZ \,.
\tag{B.1}
$$

This model has two U(1) symmetries under which the fields have the following charges,

$$
X : \ (1,0) \,, \qquad Y : \ (0,1) \,, \qquad Z : \ (-1,-1) \,.
\tag{B.2}
$$

The theory has a one dimensional conformal manifold [78] on which the two U(1) symmetries are broken but we will consider the locus with the symmetries preserved.[31] The theory has chiral operators of R-charge 2/3 ($X$, $Y$, and $Z$) and chiral operators of R-charge 4/3 ($X^2$, $Y^2$, and $Z^2$).

---

[31]The deformation taking us on the conformal manifold is $\Delta W = \tau (X^3 + Y^3 + Z^3)$.

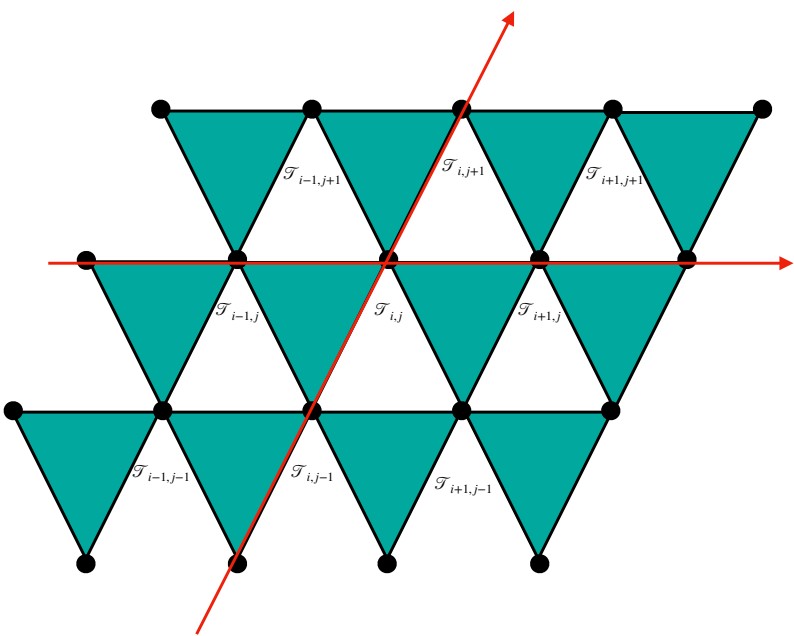

Figure 6: The two dimensional triangular lattice coupling $L_1 \times L_2$ $XYZ$ models.

Let us consider gluing copies of the $XYZ$ model into a two dimensional triangular lattice. See Figure 6. The superpotential is,

$$W = \sum_{i=1}^{L_1}\sum_{j=1}^{L_2}\Bigg( X_{i,j}Y_{i,j}Z_{i,j} + \lambda\Big( X_{i,j}(Y_{i,j+1}Z_{i+1,j} + Z_{i,j+1}Y_{i+1,j}) + Y_{i,j}(Z_{i,j+1}X_{i+1,j} + X_{i,j+1}Z_{i+1,j})$$

$$+ Z_{i,j}(X_{i,j+1}Y_{i+1,j} + Y_{i,j+1}X_{i+1,j})\Big)\Bigg). \tag{B.3}$$

The first term defines the decoupled $L_1 \times L_2$ $XYZ$ models. The rest of the superpotential couples triplets of nearest neighbor $XYZ$ models together. Each triplet has two U(1) symmetries before the gluing. There are six different interactions preserving a diagonal combination of the symmetries of each copy. Thus each gluing gives rise to $6 - 2 \times 3 + 2 = 2$ exactly marginal deformations. We choose a particular combination with coupling $\lambda$ above to have translational symmetry on the lattice.

Let us consider one of the two U(1) symmetries and write the non-conservation equation,

$$\overline{D}^2 J_{i,j}^1 = X(\lambda)\Big( X_{i,j}(Y_{i,j+1}Z_{i+1,j} + Z_{i,j+1}Y_{i+1,j}) - Z_{i,j}(X_{i,j+1}Y_{i+1,j} + Y_{i,j+1}X_{i+1,j}) \tag{B.4}$$

$$- X_{i-1,j}Y_{i-1,j+1}Z_{i,j} + Z_{i-1,j}Y_{i-1,j+1}X_{i,j} + Y_{i-1,j}(Z_{i-1,j+1}X_{i,j} - X_{i-1,j+1}Z_{i,j})$$

$$- X_{i,j-1}Z_{i,j}Y_{i+1,j-1} + Y_{i,j-1}(-Z_{i,j}X_{i+1,j-1} + X_{i,j}Z_{i+1,j-1}) + Z_{i,j-1}X_{i,j}Y_{i+1,j-1}\Big).$$

This can be written as,

$$\overline{D}^2 J_{i,j}^1 + \mathcal{J}_{\rightarrow}^1(i,j) - \mathcal{J}_{\rightarrow}^1(i-1,j) + \mathcal{J}_{\uparrow}^1(i,j) - \mathcal{J}_{\uparrow}^1(i,j-1) = 0, \tag{B.5}$$

where we have defined,

$$\mathcal{J}_{\rightarrow}^1(i,j) = X(\lambda)\Big( Z_{i,j}X_{i+1,j}Y_{i,j+1} - X_{i,j}Y_{i,j+1}Z_{i+1,j} + Y_{i,j}Z_{i,j+1}X_{i+1,j} - Y_{i,j}X_{i,j+1}Z_{i+1,j}\Big),$$

$$\mathcal{J}_{\uparrow}^1(i,j) = X(\lambda)\Big( Z_{i,j}Y_{i+1,j}X_{i,j+1} - X_{i,j}Z_{i,j+1}Y_{i+1,j} - Y_{i,j}Z_{i,j+1}X_{i+1,j} + Y_{i,j}X_{i,j+1}Z_{i+1,j}\Big). \tag{B.6}$$

The equations for the second U(1) symmetry are obtained by exchanging $X$ with $Y$ in all the expressions. We obtain for both U(1) symmetries a pair of exactly marginal operators, topological on the lattice, and corresponding to the two cycles of the lattice torus.

Let us perform a more thorough counting of nearest neighbour deformations. In total we have six marginal deformations for every gluing and thus $6 \times L_1 \times L_2$ in total. We have $2 \times L_1 \times L_2$ U(1) symmetries with the deformations preserving two of them. Thus in total the number of exactly marginal deformations is,

$$6 \times L_1 \times L_2 - 2 \times L_1 \times L_2 + 2 = 4 \times L_1 \times L_2 + 2. \tag{B.7}$$

A way to understand this is that we have $2 \times L_1 \times L_2$ exactly marginal deformations corresponding to every gluing (the grey triangles), twice $L_1 \times L_2 - 1$ deformations corresponding to the holonomies on the lattice not winding the cycles: these are the holonomies for the two U(1) symmetries,

$$\Delta W_{ij}^{1/2} = \mathcal{J}_{\to}^{1,2}(\mathbf{x}, i, j) + \mathcal{J}_{\uparrow}^{1,2}(\mathbf{x}, i+1, j) - \mathcal{J}_{\to}^{1,2}(\mathbf{x}, i, j+1) - \mathcal{J}_{\uparrow}^{1,2}(\mathbf{x}, i, j). \tag{B.8}$$

Finally we have the two topological deformations for each U(1) symmetry corresponding to the cycles.

## C  A concrete example of the $D = 6$ constructions

As a concrete example of $4pt$ sphere, let us consider $\mathfrak{a}_1$ class $\mathcal{S}$ [9,42,79] (See also [40,43,80].) Here the $4pt$ sphere is an $\mathfrak{su}(2)$ SQCD with $N_f = 4$. The symmetry $\mathfrak{g}_P$ is $\mathfrak{u}(1)$ (the Cartan of $\mathfrak{g}_{6d} = \mathfrak{su}(2)$) and the marginal operators are two chiral operators with charges $+2$ and $-2$ under this symmetry. This means that $\mathfrak{g}_p = \mathfrak{u}(1)_t$ and $\tilde{\mathfrak{g}}_P = \emptyset$. We think of this theory as gluing of two tri-fundamentals of $\mathfrak{su}(2)$, $Q$ and $Q'$, by gauging a diagonal combination of one of the puncture $\mathfrak{g}_{5d} = \mathfrak{su}(2)$. The fields $Q$ have $\mathfrak{u}(1)_t$ charge $+\frac{1}{2}$ and $Q'$ charge $-\frac{1}{2}$. The marginal operators are then $\text{Tr}(Q^2)^2$ and $\text{Tr}(Q'^2)^2$. The exactly marginal operator corresponding to the complex structure modulus is $\text{Tr} Q^2 Q'^2$. The moment map operators are $Q^2$ and $Q'^2$. These operators are in the adjoint irrep of one of the puncture $\mathfrak{su}(2)$ symmetries and have charge $\pm 1$ under the $\mathfrak{u}(1)_P$.

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
