# Peer review of "Generalized lattices, conformal manifolds, and symmetries"

_SciPost Physics, doi:SciPost Phys. 19, 069 (2025)_

## Round 1 · Referee Report · Anonymous (Referee 2) · 2025-6-9

Report

The article explores supersymmetric conformal field theories (SCFTs) constructed on a lattice, typically by coupling multiple copies of a fundamental SCFT. The authors investigate the interplay between the conformal manifolds of these theories and their global, on-site, and lattice symmetries. A key finding is that certain non-conservation equations for broken symmetries can be reinterpreted as conservation equations by incorporating the lattice dimensions. This insight leads to the identification of specific, exactly marginal deformations associated with lattice holonomies that are topological on the lattice. The authors also discuss how certain deformations added to the action can be understood as topological lattice defects. The concepts are well-motivated and supported by concrete examples. The manuscript is well-written and the results are significant. I recommend its publication after the authors address the following points.

Requested changes

  1. The discussion of the duality depicted in Figure 5 (Section 7.2) bears a strong resemblance to the "duality" (or more precisely, the continuation past infinite coupling) in the context of five-brane webs. It is known that the theory associated with a punctured sphere can be realized using such five-brane webs (see, for example, arXiv:0906.0359). The authors should comment on this potential connection. Furthermore, it would be valuable for the authors to clarify whether the superpotentials of the theory before and after applying the duality are related by Seiberg duality.

  2. A number of minor typographical errors should be corrected throughout the manuscript:

Page 2: "in in" should be "in". Page 4: "bares more resemblances" should be "bears more resemblance". Page 5: "it’s own O(2) symmetry" should be "its own O(2) symmetry". Page 8: "npdes" should be "nodes". Page 11: "the symmstry is extended" should be "the symmetry is extended". Page 11: "the the marginal superpotential" should be "the marginal superpotential". Page 13: "higer dimensional lattices" should be "higher-dimensional lattices". Page 13: "these marginal opertators" should be "these marginal operators". Page 13: "one obtaines" should be "one obtains". Page 13: "large enough value the flux" should be "large enough value of the flux". Page 16: "operators corersponding to paths" should be "operators corresponding to paths". Page 23: "several question" should be "several questions". Page 24: "it’s Cartan sub-group" should be "its Cartan subgroup".

Recommendation

Ask for minor revision

  • validity: -
  • significance: -
  • originality: -
  • clarity: -
  • formatting: -
  • grammar: -

Author:  Shlomo Razamat  on 2025-07-22  [id 5659]

(in reply to Report 2 on 2025-06-09)
Category:
remark

We are grateful to the referee for the careful reading of the manuscript and their comments.

1) As all the manipulations discussed in the paper preserve conformality, the dualities we discuss are not IR/Seiberg ones, but rather conformal. We will add a footnote stressing this point.

2) We thank the referee for suggesting to mention brane webs: indeed although our discussion is more general, in some cases the duality manipulations we mention can have brane avatars. We will add a footnote mentioning this fact.

We thank again the referee for the physics comments and catching the typos.

---

## Round 1 · Referee Report · Anonymous (Referee 1) · 2025-6-9

Strengths

  • makes the very interesting observation that SCFT's defined by a lattice constructed by joining elementary SCFT's can naturally exhibit higher-dimensional behavior in the form of current non-conservation equations being interpreted as conservation equations in higher dimensions

Report

This paper studies SCFT’s defined by lattices in a precise way. To each lattice site a SCFT is associated, and to each lattice link a marginal coupling between the corresponding SCFT’s is associated, allip in such a way that the coupling partially breaks of the global symmetry which the decoupled system would otherwise have. As the authors show, the broken conservation equations can be interpreted as conservation equations in one dimension higher, where the extra dimension is the lattice itself. Moreover, it is possible to interpret the associated conserved charges in terms of symmetry defects. The authors then analyze several examples, including the interesting (and more involved) case of compactifications of 6d SCFT’s.

In the results presented in the paper, it seems very relevant that the lattice (the quiver) is periodic. For instance, the topological charge operator in the lattice direction relies on the lattice being periodic (in that respect, would the (non)-conservation equations around eq. (11) be different for the boundaries of an open lattice?). Can open lattices can still host similar symmetries, perhaps upon adding suitable boundary conditions”?

The case in section 5.2 seems to include the Z_N orbifold of the conifold (more generically, section 6 seems to include Z_NxZ_M orbifolds of the conifold), which is a deformation of the N=2 necklace quiver briefly mentioned in footnote 20. Even though perhaps beyond the scope of the paper, it is natural to wonder about the string theory interpretation of the higher-dimensional symmetry found in the paper. In addition, given the relation to the N=2 necklace quiver, it is natural to wonder about its meaning in the context of standard deconstruction. Another natural question, also probably beyond the scope of the paper, is whether these symmetries can be gauged.

The paper presents interesting novel ideas and it is well-written. I find the paper very interesting and recommend it for publication.

Requested changes

Typos:

  • page 8, 7th line from top: "npdes" -> nodes
  • page 21, 16th line from bottom: "continues" -> continuous

Recommendation

Publish (easily meets expectations and criteria for this Journal; among top 50%)

  • validity: good
  • significance: good
  • originality: high
  • clarity: high
  • formatting: excellent
  • grammar: excellent

Author:  Shlomo Razamat  on 2025-07-22  [id 5660]

(in reply to Report 1 on 2025-06-09)
Category:
remark
question

We are grateful to the referee for their time and the thoughtful comments on the manuscript.

1) Even in presence of the boundary a symmetry is preserved and conservation equation 11 is true in the bulk of the lattice. On the boundary indeed it needs to be corrected. Whether a chain is open or closed has an important effect on counting exactly marginal operators: this is an avatar of the referees comment. This change in counting of exactly marginal operators is directly related to the conservation equation being different on the boundary. We discuss this aspect of the open chain in the fourth paragraph on page 8 and in footnote 14.

2) Indeed, at least some of the constructions we discuss have brane realizations and it would be very interesting to understand our claims in that context. In this paper, as it is purely field theoretical, we left such considerations for future work. Let us stress here that deconstruction is importantly different from what we are doing: here we insist on not breaking the conformal symmetry while performing our manipulations, while deconstruction (though starting with similar conformal lattices of QFTs) involves turning on vacuum expectation values (and thus breaking conformality) when taking the continuum limit.

3) We thank the referee for stressing the importance of understanding gauging of symmetries. Understanding various anomalies of lattice symmetries and "on site'' symmetries in such constructions is one of our main goals for extending this research.

We would like to thank the referee for the interesting comments.

---

## Round 2 · Author Response

We have addressed the referees comments and corrected several typos.

---

## Round 2 · List of Changes

1) Footnote 27 was added to stress that the dualities we consider are conformal dualities.

2) Footnote 28 was added to mention a possible connection to brane constructions.

3) The spelling and grammar typos caught by the referees were corrected.

---

## Editorial Decision

published